# Epidemiology and Molecular Analyses of Influenza B Viruses in Senegal from 2010 to 2019

**DOI:** 10.3390/v14051063

**Published:** 2022-05-16

**Authors:** Cheikh Talibouya Touré, Amary Fall, Soa Fy Andriamandimby, Mamadou Malado Jallow, Deborah Goudiaby, Davy Kiori, Sara Sy, Yague Diaw, Koba Ndiende Ndiaye, Fatimata Mbaye, Mbayang Ndiaye Niang, Jean Michel Heraud, Ndongo Dia

**Affiliations:** 1Virology Department, Institute Pasteur de Dakar, Dakar 12000, Senegal; cheikhtalibouya.toure04@gmail.com (C.T.T.); afall2@jhmi.edu (A.F.); mamadoumaladojallow@gmail.com (M.M.J.); deborah.goudiaby@pasteur.sn (D.G.); davy.kiori@pasteur.sn (D.K.); sara.sy@pasteur.sn (S.S.); yague.diaw@pasteur.sn (Y.D.); ndiendekoban@gmail.com (K.N.N.); mbndiaye2002@gmail.com (M.N.N.); jean-michel.heraud@pasteur.fr (J.M.H.); 2Department of Animal Biology, Faculty of Science, University Cheikh Anta Diop, Dakar 12000, Senegal; fatimata2.mbaye@ucad.edu.sn; 3Virology Unit, Institute Pasteur de Madagascar, Antananarivo 101, Madagascar; soafy@pasteur.mg

**Keywords:** influenza B, surveillance, epidemiology, phylogenetic analyses, reassorting viruses, Senegal, Africa

## Abstract

Influenza virus types A and B are responsible for acute viral infections that affect annually 1 billion people, with 290,000 to 650,000 deaths worldwide. In this study, we investigated the circulation of influenza B viruses over a 10-year period (2010–2019). Specimens from patients suspected of influenza infection were collected. Influenza detection was performed following RNA extraction and real-time RT-PCR. Genes coding for hemagglutinin (HA) and neuraminidase (NA) of influenza B viruses were partially sequenced, and phylogenetic analyses were carried out subsequently. During the study period, we received and tested a total of 15,156 specimens. Influenza B virus was detected in 1322 (8.7%) specimens. The mean age of influenza B positive patients was 10.9 years. When compared to reference viruses, HA genes from Senegalese circulating viruses showed deletions in the HA1 region. Phylogenetic analysis highlighted the co-circulation of B/Victoria and B/Yamagata lineage viruses with reassortant viruses. We also noted a clear seasonal pattern of circulation of influenza B viruses in Senegal.

## 1. Introduction

Influenza viruses are contagious pathogens which cause acute viral infections affecting around 1 billion individuals per year, with 3 to 5 million severe cases and 290,000 to 650,000 deaths worldwide [1]. Influenza disease burden is particularly highest in countries with limited resources, mainly located in tropical areas [2]. Amongst different types of influenza viruses, around 30% of the morbidity and mortality associated with influenza is attributed to influenza B (IBV) [3,4]. Until recently, IBVs were thought to cause milder illness compared to influenza A viruses (IAV), but several studies have showed that infections with IAV or IBV are clinically indistinguishable, and can lead to severe complications in both children and adults [3,5,6].

Influenza viruses belong to the family *Orthomyxoviridae*, which consists of segmented, single-stranded negative sense RNA viruses. They are classified into four types: A, B, C and D [7,8]. While IAV subtypes depend on the combination of the hemagglutinin (HA) and the neuraminidase (NA) genes, IBVs are divided into two lineages based on their HA gene: the Victoria lineage (B/Vic) and the Yamagata lineage (B/Yam) [9]. It is estimated that these two lineages genetically diverged in around 1983 [10]. In the 1990s, the B/Yam lineage was the predominant virus worldwide, while B/Vic viruses circulated mainly in East Asia. B/Vic viruses re-emerged in North America and in Europe in 2001, and therefore spread globally [11,12]. Since then, these two B lineages are co-circulating worldwide, with an alternating dominance of each lineage from year to year [13].

Despite its public health importance, IBV is poorly studied compared to IAV, especially in Africa. Our study aimed at investigating the circulation of IBV over a 10-year period (2010–2019) in the context of the influenza surveillance conducted in Senegal. We documented the genetic diversity and evolutionary dynamics of IBV during the study period.

## 2. Materials and Methods

### 2.1. Clinical Specimens’ Collection

From 2010 to 2019, we collected respiratory specimens, as well as epidemiologic data, as part of the sentinel influenza surveillance conducted in Senegal by the National Influenza Centre (NIC) hosted at the Institute Pasteur de Dakar (IPD). Nasopharyngeal and/or Oropharyngeal swabs were collected from outpatients who met the WHO case definition of influenza-like illness (ILI), or from hospitalized patients with severe acute respiratory infection (SARI) as previously described [14,15]. Swabs were placed into universal viral transport medium (Becton Dickinson, Italy), stored at 4–8 °C and transported to NIC within 72 h of collection for molecular detection of influenza viruses. One aliquot of each sample was stored at −80 °C for biobanking and further studies.

### 2.2. RNA Extraction and Real Time Reverse Transcription-PCR (qRT-PCR)

The nucleic acids purification from respiratory specimens was done using the QIAGEN Kit (QIAamp Viral RNA Mini Kit, Germany), with a matrix volume of 200 µL and a final elution volume of 60 µL, according to the manufacturer’s instructions and as previously described [16]. RNA samples were tested for IAV and IBV using the human influenza rRT-PCR diagnostic panel, and the influenza B lineage detection kit, kindly provided by the US Centers for Disease Control and Prevention (US CDC). Amplifications were performed using the AgPath-ID TM One-Step rRT-PCR kit (Termo Fisher Scientifc, Austin, TX, USA), according to the manufacturer’s instructions. After a screening of influenza viruses, a selection of IBV positive specimens was carried out for molecular characterization. Specimens selected for sequencing were representative of the entire study period, and had Ct values equal to or lower than 30.

### 2.3. HA/NA Genes Amplification and Sequencing

HA and NA genes of selected positive IBV samples were amplified using the Quick-Load One Taq One-Step Kit (New England Biolabs, Ipswich, MA, USA). The primers INFB-HA-F1 (5′ATGAAGGCAATAATTGTACTACTC-3′) and INFB-HA-R3 (5′-CCTTATAGACAGATGGAGCAAG-3′) were used to amplify the whole HA gene (1756 bp fragment long). Amplification of the NA gene (1429 pb) was done using the INFB-NA-F1 (5’-TGAACAATGCTACCTTCAAC-3′) and INFB-NA-R2 (5′-GAACAGAYTCAACCATTCCT CC-3′) primers. In a final volume of 33 µL, we used 14 µL of the reaction mix to which we added 0.5 µL of enzyme, 1 µL of forward primer, 1 µL of reverse primer, 6.5 µL of nuclease free water and 10 µL of viral RNA. The RT-PCR was performed with a reverse transcription phase at 48 °C for 30 s; an initial denaturation step at 94 °C for 1 min; and 45 cycles comprising a denaturation phase at 94 °C for 15 s, an annealing phase at 50 °C for 30 s, and an extension phase at 68 °C for 2 min. A final extension phase was performed at 68 °C for 5 min. Amplicons obtained with the positive samples were purified with the MACHEREY-NAGEL kit (NucleoSpin Gel and PCR Clean-UP, Macherey-Nagel, Düren, Germany). Purified products were sent for bidirectional sequencing to Genewiz Service (Bodman-Ludwigshafen, Germany).

### 2.4. Phylogenetic Analysis

Phylogenetic analyses were carried out by adding sequences from other African countries strains downloaded from the Global Initiative on Sharing All Influenza Data (GISAID) platform, as well as adding sequences of reference strains for clades. The sequences obtained in Fasta format were cleaned with Bioedit version 7.2.5 [17]. Alignment of all sequences was performed using Bioedit version 7.2.5. The phylogenetic trees for HA and NA genes were generated using the maximum likelihood (ML) method using IIQ-TREE version 1.6.12 [18], and the visualization was done with FigTree version 1.4.4 [19]. Robustness of trees topology was accessed with 1000 replicates, and bootstrap values greater than 70% were shown on branches of the consensus trees.

### 2.5. Phylodynamic Analysis

The time-scaled phylogeny of influenza B for both HA and NA genes was estimated using the Markov Chain Monte Carlo (MCMC) method implemented in BEAST v1.10.4 [20]. The stick clock was selected as the best model found. Bayesian Skyline age coalescent tree priors were used to calculate the time to the most recent common ancestor (tMRCA) [21,22]. The GTR+G substitution model was selected as the best model using MEGA software version 7.0.26 [23]. The MCMC chain was run for 200,000,000 generations, with sampling every 2000 steps for achieved convergence. Tree Annotator v1.10.4 generated a maximum clade credibility (MCC) tree after removing the first 10% of trees as burn-in. The FigTree v1.4.4 [19] program was used for visualization of the phylogenetic tree with genotypes and divergence time scale.

### 2.6. Aligned Amino Acid Sequence Analysis

The amino acid sequences alignment was performed using MEGA software version 7.0.26 [23] to detect the mutations present in the targeted genes.

### 2.7. Statistical Analyses

Statistical analyses were performed using R software version 4.0.3 [24]. Categorical variables are reported as counts and percentages. Continuous variables were expressed with their 95% Confidence interval (CI 95%). Comparisons were conducted using a chi-square test. For the demographic and clinical data of all patients and those with confirmed IBV infections, the percentages were compared using a Chi-square test, and a *p*-value < 0.05 was considered statistically significant. The degree of association between outcome variables was expressed as an odds ratio (OR), taking account of potential confounding factors.

## 3. Results

### 3.1. Demographic and Clinical Characteristics of Patients

A total of 15,156 patients were sampled and tested for influenza viruses between 2010 and 2019 (Table 1). The male to female ratio was 0.99 (7560/7596). The age of patients ranged from 3 days to 99 years, with a median age of 4.6 years (Interquartile Range (IQR): 1.6–15.5) and a mean age of 11.4 years (CI 95%: [11.1–11.6]). Children under 5 years represented the main age group (52.2%) of patients sampled, followed by age group 5–15 years (22.2%). The most reported symptoms in patients were fever (92.6%) and cough (81.9%), followed by rhinorrhea (42.3%), headache (17.1%), sore throat (15.8%) and myalgia (13.9%).

### 3.2. Characteristics of Patients Infected with Influenza Viruses

Of the 15,156 specimens received, 5408 were tested positive for influenza. Among positive patients, the sex ratio M/F was 0.91 (2581/2827). We noted that women were more likely to be infected with influenza than men, with a crude odds ratio OR = 1.14 (*p* < 0.001). When compared to children under 5 years old, multivariate analyses showed that patients aged 5 years and more were more likely to be infected with influenza with OR ranging from 1.63 to 1.66 (*p* < 0.001). Regarding symptoms, on multivariate analyses we found that fever, cough, rhinorrhea, headache and myalgia were positively associated with influenza infection (*p* < 0.001), while vomiting and conjunctivitis were less likely to be associated with influenza infection (*p* < 0.001).

### 3.3. Characteristics of Patients According the Type of Infection (IAV vs. IBV)

Of the 5408 patients who tested positive for influenza, 5315 were infected with only one virus, and 93 patients had coinfection (IAV and IBV). A total of 3993 (75%) patients were infected with IAV, while 1322 (25%) were infected with IBV (Table 2). We analyzed data of patients with only one infection and compared their clinical characteristics (Table 2). No difference was found between IAV and IBV-infected patients according to sex. The mean age of patients infected with IAV was statistically higher than that of patients infected with IBV, with the ages being respectively 13.3 [95% CI: 12.8–13.8] years and 10.2 [95% CI: 9.6–10.9] years (*p* < 0.001). When compared to age group [0–5] years, age group [5–15] years was more likely to be infected with IBV, while groups aged 15 years and older were more likely to be infected with IAV (*p* < 0.001). Regarding symptoms of influenza-positive patients, multivariate analyses showed that fever was more frequent among IAV infected patients while cough and rhinorrhea were more frequently associated with IBV infection. No difference was noted for other symptoms.

### 3.4. Pattern of Circulation of Influenza Viruses in Senegal from 2010 to 2019

During the study period, we observed that IAV and IBV have circulated every year but with various intensities (Figure 1). Overall, IAV was the most prevalent virus detected, except in 2010, 2011 and 2017 where the prevalence of IAV and IBV were similar. We implemented B-lineages detection since 2015, and from 2015 to 2019, we observed that B/Yam was the dominant strain only in 2015. In 2016, almost no circulation of either B lineage was observed. From 2017 to 2019, we noted a co-circulation of the two lineages, but B/Vic remained the dominant strain.

### 3.5. Amino Acid Sequences Analysis of IBV Strains Circulating in Senegal

Alignment and comparison of Senegalese and reference (FJ766842.1) sequences showed deletions (Δ) in the HA1 region of several sequences of B/Vic viruses that circulated in Senegal between 2018 and 2019. These sequences were characterized by a loss of three amino acids at positions 162 to 164 (D162, K163, N164) (Appendix A).

### 3.6. Phylogenetic Analyses of IVB Strains Circulating in Senegal, 2010–2019

A total of 108 and 89 sequences for respectively HA and NA genes were successfully generated from IBV-positive samples. As previously observed in lineages detection, the HA phylogeny showed co-circulation of the two IBV lineages, with a predominance of the B/Vic lineage (69.9%; 68/108) (Figure 2). Inside the B/Vic lineage, we identified two clades: the minority clade 1B (2/68) which was exclusively detected in 2010, and the clade 1A (66/68) which circulated from 2012 to 2019. Regarding B/Yam, Senegalese strains clustered with clades 2 and 3, with a high predominance of the B/Yam clade 3 (24/29). We also observed that the B/Yam clade 2 viruses were exclusively detected in 2013, while B/Yam clade 3 circulated in 2014.

Interestingly, analysis of the NA showed that some IBV strains appeared to be reassortants between B/Yam and B/Vic viruses (Figure 3). Indeed, twelve (eleven in 2015 and one in 2016) strains belonging to the B/Yam clade 3 were based on the HA tree, and clustered with the B/Vic clade 1A based on the NA gene phylogeny. We also detected one strain in 2018 that belonged to B/Vic lineage based on the HA gene, and to the B/Yam lineage based on the NA gene. In addition, we observed intra-lineage reassortment cases between B/Vic strains. Indeed, seven (six in 2013 and one in 2012) reassortants of the B/Vic clade 1A possessed an NA gene from the B/Vic clade 6.

### 3.7. Bayesian Dated Tree Estimates

Markov Chain Monte Carlo (MCMC) algorithm was generated to compute the time-scaled evolutionary relationship, in order to estimate the mean evolutionary and substitution rates of IBV genotypes from 2010 to 2019. The time to the most recent common ancestor (tMRCA) of IVB was estimated at around 1970 (95% highest posterior density (HPD): 1968–1975) using the HA gene (Appendix A), and around 1980 (95% HPD: 1977–1983) using the NA gene (Appendix A). The dominant clade 1A of the B/Vic lineage emerged in 2010 (95% HPD: 2009–2011) in Senegal. Viruses characterized by Δ161–163 in the HA1 region emerged in 2017 (95% HPD: 2017–2018). The B/Yam lineage clade 2 emerged in 2012 (95% HPD: 2012–2013), and clade 3 in 2010 (95% HPD: 2010–2011). For reassortant strains, those having undergone a Yamagata HA gene and Victoria NA gene reassortment emerged in 2014 (95% HPD: 2014–2015). For those with the Victoria HA gene and the Yamagata NA gene, emergence was dated to 2018 (95% HPD: 2017–2018). Reassortant strains within the Victoria lineage (clade 1A and clade 6) emerged in 2011 (95% HPD: 2010–2011).

## 4. Discussion

The present study focused on the epidemiology and evolutionary dynamics of influenza type B viruses that circulated in Senegal from 2010 to 2019. On average, IBV accounted for approximately 25% of influenza-positive cases during our study period. This rate is similar to those found in England (25%: 2003–2013), Uruguay (26%: 2012–2019) and Finland (26%: 1999–2012); however, this rate was higher than the worldwide proportion (20%) reported in many countries from different geographical areas, including South Africa (17%: 2005–2014), Australia (17%: 2001–2014), the United States of America (16%: 2000–2012) and Brazil (19%: 2006–2014) [3,25,26,27]. However, higher rates were found in China (Southern China (29%: 2006–2012 and 2011–2017); Northern China (31%: 2005–2012)) [25,28], Kenya (31%: 2012–2016) [29] and Madagascar (33%: 2002–2013) [25]. Among patients presenting ILI or SARI symptoms during the study period, our data showed a global prevalence of 8.6% of influenza B. This prevalence observed is quite similar to those recorded in Malaysia (9%) [30] and in Myanmar 10% [31]. However, a lower prevalence of IBV was reported in other countries, such as India (4%) [32], Ethiopia and Madagascar with 6% [33,34], China (5%) [35] and Uruguay (4%) [36].

Regarding the detection rate of IBV per age, even if the virus was detected in all age groups, the vast majority (78%; 1037/1322) of cases were detected in the pediatric population (less than 15 years old). The same results have been observed in several other studies carried out in Australia (63%), Finland (52%) and Thailand (67%) [26,37,38]. It is believed that young children are more at risk of getting infected with influenza compare to older ages, as a result of a naïve immune system in early childhood [39]. Although children aged less than 5 years represented 43% of all IBV-infected patients, age group [5,6,7,8,9,10,11,12,13,14,15] years was more likely to be infected with IBV. Nevertheless, these data could reflect a bias in our recruitment, since our study population was mainly pediatric.

The amino acid sequence alignments of the HA gene showed a deletion of three amino acids in the Victoria lineage. Similar strains were detected in different regions, and they clustered with the B/Washington/02/2019 virus belonging to the B/Vic 1A subclade [40,41]. It seems that this strain with Δ162–164 (D162, K163, N164) replaced the older strains that circulated in Senegal in previous seasons. It is to be noted that Δ163–164 (K163 and N164) in the HA1 subunit defined a new antigenic subgroup represented by B/Norway/2409/2017. These viruses have been detected in several countries [42]. B/Vic viruses circulating in Senegal with Δ162–164 (D162, K163 and N164) belonged to clades 1A.2–1A.4. Based on antigenic tests performed at the Francis Crick Institute (WHO Collaborative Centre in London), it was observed that these isolates were not recognized by some antisera directed against B/Norway/2409/2017 of Clade 1A, which could have impacted the efficacy of vaccine-containing B/Colorado/06/2017-like virus, representing the emergent strain of B/Victoria-lineage viruses with Δ163–164 deletion in HA1. However, no particular pathogenicity was reported thus far.

The phylogenetic analysis and IBV lineages detection showed a co-circulation of B/Vic and B/Yam lineages in Senegal, with a predominance of the B/Vic lineage. Similar observations were noted in other countries, such as Australia and New Zealand [43], Kenya [44], Italy [45] and Cameroon [46]. Analysis of the HA phylogenetic tree showed the circulation of two Yamagata lineage clades (clade 2 and 3) and two Victoria lineage clades (clade 1A and 1B) in Senegal from 2010 to 2019. We also observed that the clade 1A of B/Vic and the clade 3 of B/Yam were the only circulating strains in Senegal since 2012 and 2015, respectively. As observed in Senegal, circulations of B/Vic (clade 1A and 1B) and B/Yam (clades 2 and 3) viruses were reported in several countries, including Kenya [44], Thailand [38], Italy [45], Uruguay [36], Malaysia [30], China [28] and Cameroon [46]. Nonetheless, the shift in patterns of dominance of the clades can differ from one country to another. Interestingly, we observed B/Yam-B/Victoria inter-lineage reassortment (Yamagata-lineage HA and Victoria-lineage NA) in 2015 and 2016. These types of reassortments have also been reported in several countries, such as Thailand in 2014–2015 [47], Cameroon between 2014 and 2017 [46], China in 2014–2015 season [28] and Kenya in 2016 [44]. In addition, analyses of sequences from GISAID showed that Yamagata clade 3 reassortant viruses were circulating in several other African countries, including Burkina Faso (2015), Congo (2016), Côte d’Ivoire (2015), Ghana (2015–2016), Mali (2015–2016), Mozambique (2015), Niger (2015–2017), Nigeria (2015), Rwanda (2016), South Africa (2015), Tanzania (2015–2016), Togo (2016), Uganda (2016) and Zambia (2014–2015). In addition to these inter-lineage reassortments, we noted some cases of intra-lineage reassortments in 2012 and 2013, with Victoria lineage clade 1A reassortants possessing a Victoria lineage clade 6 NA segment. This reassortment was observed in many regions of the world, and was noted as B/Dakar/10/2012-like NA clade [13]. Nonetheless, it has been shown that reassortment is one of the evolution mechanisms of IBV, allowing it to adapt to selection pressure [48].

The time-scaled maximum clade credibility (MCC) tree suggested that the tMRCA of IBV was estimated at around 1970 (95% HPD: 1968–1975) using the HA gene, and at around 1980 (95% HPD: 1977–1983) using the NA gene. This finding is in line with a study carried out in Kenya [44].

## 5. Conclusions

The present study highlights the epidemiology and evolution dynamics of IBV strains that circulated in Senegal between 2010 and 2019. IBV activity systematically peaked between August and October, and mainly affected young aged 5 to 15 years old. More studies are needed to estimate the disease burden, as well as the severity associated with influenza infections in Senegal, in different age groups and populations. During our study, we detected inter-lineage as well as intra-lineage reassortants. It would be interesting to assess the impact of such reassortments on the disease presentation and transmission rate of these viruses. Our study emphasizes the need to develop genomic surveillance of influenza viruses, in order to detect new emerging strains that could threaten public health.

## Figures and Tables

**Figure 1 viruses-14-01063-f001:**
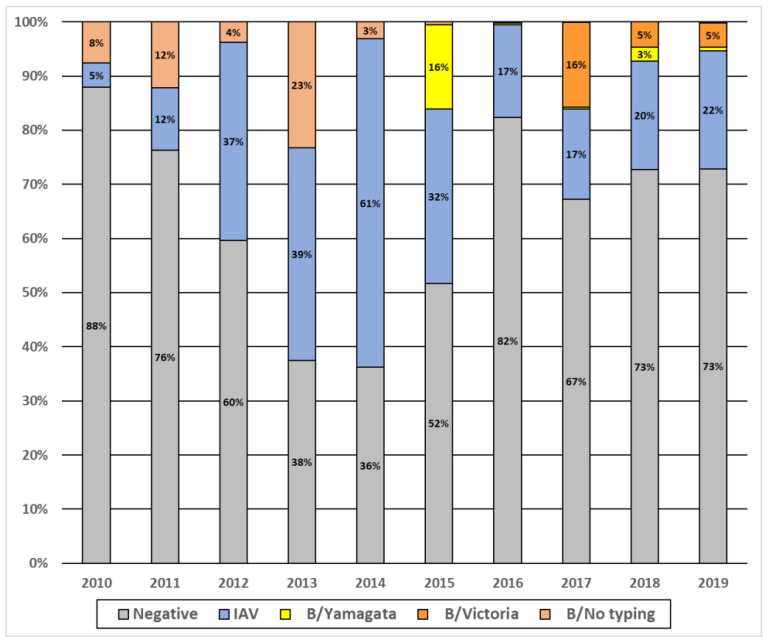
Annual prevalence of influenza viruses A and B in Senegal, 2010–2019.

**Figure 2 viruses-14-01063-f002:**
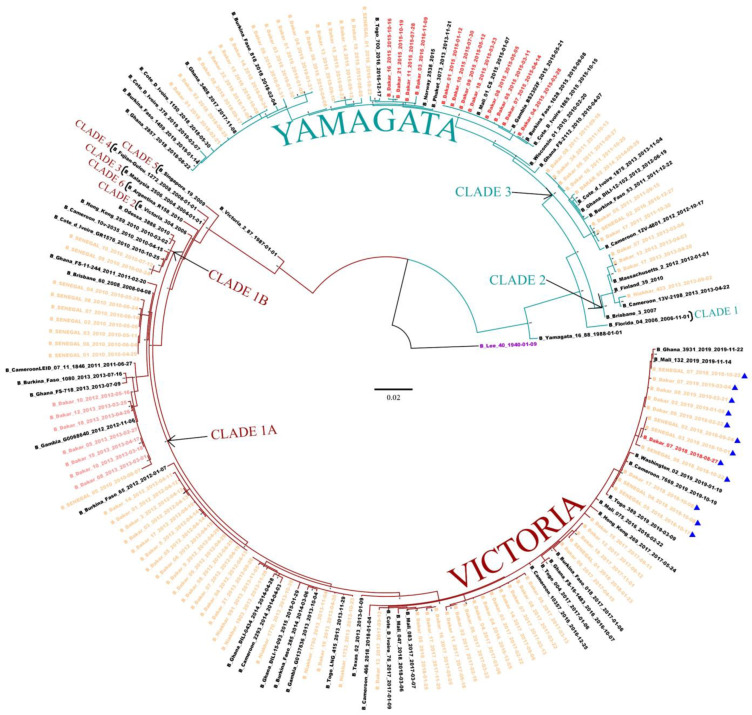
The maximum likelihood (ML) tree for the HA gene of IBV circulating in Senegal. ML tree was calculated using the IQ-TREE software version 1.6.12 [18] and visualized using the Figtree software version 1.4.4 [19]. A bootstrap number of 1000 was used, and the software was responsible for defining the correct model used. Strain B/Lee/1940 was used in order to root the tree (purple). Senegalese strains, inter-lineage recombinant strains, and intra-lineage recombinant stains are respectively indicated in orange, red and pink. Strains with Δ162–164 in the HA gene (D162, K163, N164) are also indicated (▲).

**Figure 3 viruses-14-01063-f003:**
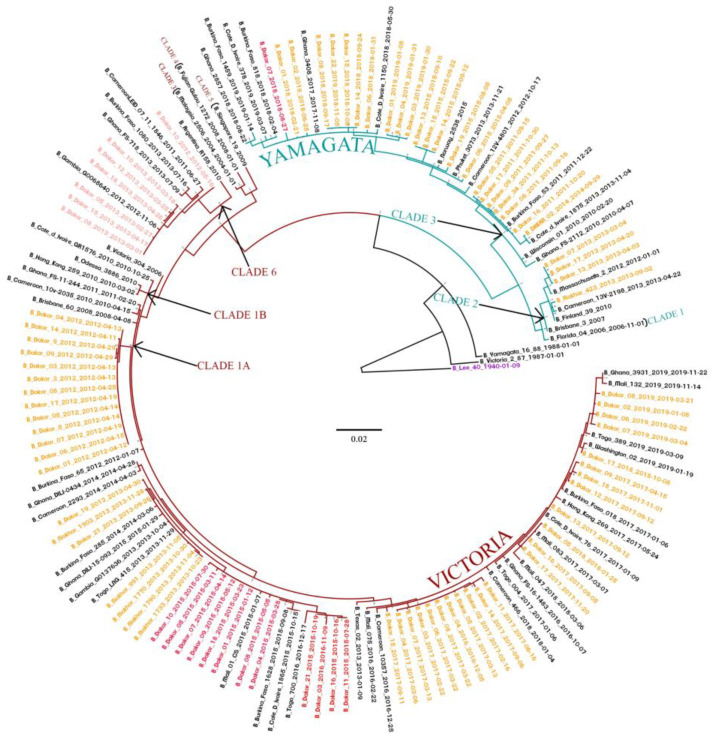
The maximum likelihood (ML) tree for the NA gene of IBV circulating in Senegal. The ML tree was calculated using the IQ-TREE software version 1.6.12 [18] and visualized using the Figtree software version 1.4.4 [19]. A bootstrap number of 1000 was used, and the software was responsible for defining the correct model used. Strain B/Lee/1940 was used in order to root the tree. Senegalese strains, inter-lineage recombinant strains, and intra-lineage recombinant stains are respectively indicated in orange, red and pink.

**Table 1 viruses-14-01063-t001:** Characteristics of patients presenting ILI and or SARI symptoms according to influenza infection. Senegal 2010–2019.

	Total (N = 15,156)	Positive (*n* = 5408)	Negative (*n* = 9748)	*p*-Value	Crude OR (95% CI)	Adjusted OR (95% CI)
**Gender *n* (%)**						
Female	7596 (50.1%)	2827 (52.3%)	4769 (48.9%)	<0.001	1.14 [1.07–1.22]	
Male	7560 (49.9%)	2581 (47.7%)	4979 (51.1%)		1	
**Age Group *n* (%)**				<0.001		
[0–5[	7907 (52.2%)	2396 (44.3%)	5511 (56.5%)		1	1
[5–15[	3372 (22.2%)	1398 (25.9%)	1974 (20.3%)		1.63 (1.50–1.77)	1.73 (1.58–1.89)
[15–25[	1520 (10.0%)	631 (11.7%)	889 (9.1%)		1.63 (1.46–1.83)	1.77 (1.56–2.01)
[25–50[	1772 (11.7%)	744 (13.8%)	1028 (10.5%)		1.66 (1.50–1.85)	1.81 (1.60–2.04)
50+	585 (3.9%)	239 (4.4%)	346 (3.5%)		1.59 (1.34–1.89)	1.69 (1.40–2.02)
**Symptoms *n* (%)**						
Fever	14,027 (92.6%)	5046 (93.3%)	8981 (92.1%)	0.009	1.19 (1.05–1.36)	1.24 (1.09–1.42)
Cough	12,408 (81.9%)	4562 (84.4%)	7846 (80.5%)	<0.001	1.31 (1.20–1.43)	1.36 (1.24–1.49)
Rhinorrhea	6416 (42.3%)	2431 (45.0%)	3985 (40.9%)	<0.001	1.18 (1.10–1.26)	1.28 (1.19–1.38)
Headache	2585 (17.1%)	1120 (20.7%)	1465 (15.0%)	<0.001	1.48 (1.35–1.61)	1.25 (1.13–1.38)
Sore throat	2401 (15.8%)	806 (14.9%)	1595 (16.4%)	0.020	0.90 (0.82–0.98)	0.71 (0.64–0.78)
Myalgia	2110 (13.9%)	900 (16.6%)	1210 (12.4%)	<0.001	1.41 (1.28–1.55)	1.41 (1.28–1.55)
Vomiting	1182 (7.8%)	362 (6.7%)	820 (8.4%)	<0.001	0.78 (0.69–0.89)	0.79 (0.70–0.91)
Dyspnea	333 (2.2%)	129 (2.4%)	204 (2.1%)	0.263	1.14 (0.91–1.43)	-
Arthralgia	290 (1.9%)	116 (2.1%)	174 (1.8%)	0.137	1.21 (0.95–1.53)	0.79 (0.61–1.01)
Conjunctivitis	113 (0.7%)	20 (0.4%)	93 (1.0%)	<0.001	0.39 (0.23–0.61)	0.41 (0.25–0.66)

**Table 2 viruses-14-01063-t002:** Characteristics of patients according the type of influenza virus detected. Senegal 2010–2019.

	All Influenza (N = 5315)	Influenza A (*n* = 3993)	Influenza B (*n* = 1322)	*p*-Value	Crude OR (95% CI)	Adjusted OR (95% CI)
**Gender *n* (%)**						
Female	2773 (52.2%)	2100 (52.6%)	673 (50.9%)	0.303	1	
Male	2542 (47.8%)	1893 (47.4%)	649 (49.1%)		0.93 (0.83–1.06)	
**Age Group *n* (%)**				<0.001		
[0–5[	2358 (44.4%)	1789 (44.8%)	569 (43.0%)		1	
[5–15[	1380 (26.0%)	912 (22.8%)	468 (35.4%)		0.62 (0.54–0.72)	0.56 (0.48–0.65)
[15–25[	619 (11.6%)	491 (12.3%)	128 (9.7%)		1.22 (0.99–1.52)	1.13 (0.91–1.41)
[25–50[	726 (13.7%)	603 (15.1%)	123 (9.3%)		1.56 (1.26–1.94)	1.58 (1.27–1.98)
50+	232 (4.4%)	198 (5.0%)	34 (2.6%)		1.85 (1.29–2.74)	1.88 (1.30–2.79)
*Mean age (95% CI)*	*11.0 (10.8–11.2)*	*13.3 (12.8–13.8)*	*10.2 (9.6–10.9)*	<0.001		
**Symptoms *n* (%)**						
Fever	4961 (93.3%)	3775 (94.5%)	1186 (89.7%)	<0.001	1.99 (1.58–2.48)	2.08 (1.65–2.62)
Cough	4484 (84.4%)	3315 (83.0%)	1169 (88.4%)	<0.001	0.64 (0.53–0.77)	0.67 (0.55–0.81)
Rhinorrhea	2395 (45.1%)	1702 (42.6%)	693 (52.4%)	<0.001	0.67 (0.60–0.76)	0.71 (0.62–0.81)
Headache	1106 (20.8%)	850 (21.3%)	256 (19.4%)	0.146	1.13 (0.96–1.32)	-
Sore throat	791 (14.9%)	613 (15.4%)	178 (13.5%)	0.104	1.17 (0.98–1.40)	-
Myalgia	874 (16.4%)	680 (17.0%)	194 (14.7%)	0.050	1.19 (1.01–1.42)	-
Vomiting	351 (6.6%)	265 (6.6%)	86 (6.5%)	0.918	1.02 (0.80–1.32)	-
Dyspnea	124 (2.3%)	93 (2.3%)	31 (2.3%)	1.000	0.99 (0.67–1.52)	-
Arthralgia	116 (2.2%)	87 (2.2%)	29 (2.2%)	1.000	0.99 (0.66–1.54)	0.64 (0.41–1.01)
Conjunctivitis	20 (0.4%)	16 (0.4%)	4 (0.3%)	0.797	1.33 (0.49–4.63)	-

## Data Availability

Not applicable.

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
