# Peer review of "Epidemiology and Molecular Analyses of Influenza B Viruses in Senegal from 2010 to 2019"

_viruses, 2022, doi:10.3390/v14051063_

Round 1
Reviewer 1 Report
Touré, et al.
“Epidemiology and molecular analyses of influenza B viruses in Senegal from 2010 to 2019”
In this manuscript, the authors investigated the epidemiology and genetic features of influenza B viruses in Senegal from 2010 to 2019 using specimens from patients suspected of influenza. Overall analyses were okay, but this manuscript should be modified in detail in several parts, as pointed below.
Points to be modified
Abstract, lines 19-20. “we received and tested a total of 17,276 specimens. Influenza B virus was detected in 1,397 (8.1%) specimens”. There are no these numbers in the manuscript.
Throughout the text, “subtypes” or “subtyping” is not used for influenza B viruses.
Line 131. Please spell out “IQR”.
Table 2. According to the text, the numbers of influenza A and B viruses are 3,993 and 1,322, respectively.
Lines 168-169. “2010” should also be excluded based on Figure 1.
Line 170. “in 2015” should be “since 2015”.
Line 185. “Figure 1” should be deleted.
Page 7. “Figure 3” should be “Figure 2”. Also, please indicate a group composed of strains with a 3-amino acid deletion in HA by the arrow, like for CLADE 1A, CLADE 1B, and so on, in this figure.
Page 8. “Figure 4” should be “Figure 3”.
Line 203. “one in 2012”. This reviewer could not find it in Figure.
Line 244. “73%”. How did the authors calculate this number?
Lines 246-248. “Although it is ….. from our data”. If the authors would compare the age group of 0-15 with other groups, it would seem that they see the hypothesis in their data.
Line 251. “13.2%”. How did the authors calculate this number?
Lines 253-254. “The annual circulation …… between August and October of each year”. There are no results about this in the manuscript.
Line 262. “this deleted strain”. This is not a suitable word, so it should be modified.
Line 270. “2015” should be “2011” based on Supplementary Figure 2.
References. Please carefully re-check references because some were cited twice.
Author Response
Reviewer 1: In this manuscript, the authors investigated the epidemiology and genetic features of influenza B viruses in Senegal from 2010 to 2019 using specimens from patients suspected of influenza. Overall analyses were okay, but this manuscript should be modified in detail in several parts, as pointed below.
Author’s response (AR): Thank you for the great interest you showed for this work, and the positive evaluation. I also appreciate your useful minor suggestions which will undoubtedly help to improve the quality of this paper.
Points to be modified
- Abstract, lines 19-20. “we received and tested a total of 17,276 specimens. Influenza B virus was detected in 1,397 (8.1%) specimens”. There are no these numbers in the manuscript.
(AR): The reviewer is right; we forgot to make changes in the abstract after a second round of data analysis. Corrections were made in this version.
- Throughout the text, “subtypes” or “subtyping” is not used for influenza B viruses.
(AR): The reviewer is right, and we corrected the misused of subtypes or subtyping for IBV and replaced it with lineages or lineages detection.
- Line 131. Please spell out “IQR”.
(AR): Done as suggested.
- Table 2. According to the text, the numbers of influenza A and B viruses are 3,993 and 1,322, respectively.
(AR): The reviewer is right, as it was a mistake in the head title of columns. We corrected Table 2 accordingly.
- Lines 168-169. “2010” should also be excluded based on Figure 1.
(AR): Thank you for drawing our attention on this mistake. Indeed, as stated in this version IBV was predominant in 2010.
- Line 170. “in 2015” should be “since 2015”.
(AR): Corrected.
- Line 185. “Figure 1” should be deleted.
(AR): Corrected.
- Page 7. “Figure 3” should be “Figure 2”. Also, please indicate a group composed of strains with a 3-amino acid deletion in HA by the arrow, like for CLADE 1A, CLADE 1B, and so on, in this figure.
(AR): Indeed, mistakes in figures numbering. Figure’s number corrected in this version and all strains with amino acid deletion were indicated in Figure 2.
- Page 8. “Figure 4” should be “Figure 3”.
(AR): Corrected.
- Line 203. “one in 2012”. This reviewer could not find it in Figure.
(AR): The 2012 reassorting strain is indeed located in Figure 2 (Clade 1A) and Figure 3 (Clade 6) namely, B_Dakar_10_2012_2012-05-16.
- Line 244. “73%”. How did the authors calculate this number?
(AR): We thank the reviewer for catching this mistake. Indeed, IBV was mainly detected in 1,037 pediatric patients (aged less than 15 years) out of 1,322 total patient representing 78.4% instead of 73%. The sentence was corrected accordingly.
- Lines 246-248. “Although it is ….. from our data”. If the authors would compare the age group of 0-15 with other groups, it would seem that they see the hypothesis in their data.
- Line 251. “13.2%”. How did the authors calculate this number?
(AR): We admitted that sentences from lines 246 to 251 were confusing. We have revised this paragraph for a better understanding as followed:
“It is believed that young children are more at risk of getting infected with influenza compare to older ages, due to a naïve immune system in early childhood [40]. Although children aged less than 5 years represented 43% of all IBV infected patients, age group [5-15[ years were more likely to be infected with IBV. Nevertheless, these data could be a bias in our recruitment since our study population was mainly pediatric.”
- Lines 253-254. “The annual circulation …… between August and October of each year”. There are no results about this in the manuscript.
(AR): We agree with the reviewer and since we did not present these data in our manuscript, we decided to remove this paragraph.
- Line 262. “this deleted strain”. This is not a suitable word, so it should be modified.
(AR): The sentence is rephrased.
- Line 270. “2015” should be “2011” based on Supplementary Figure 2.
(AR): As seen in Figure 3, B/Yam-B/Victoria inter-lineage reassortment (red strains) was observed in 2015 and 2016 and we postulated that these strains emerged in 2014
- Please carefully re-check references because some were cited twice.
(AR): All references have been checked and revised when needed
Reviewer 2 Report
This paper introduces the epidemic circulation and cases analysis of influenza B in Senegal from 2010 to 2019. The sample size is abundant. The data is very impotant for guiding the prevention and control of B subtype influenza for Senegal and Africa. However, the manucrispt should be improved as follows.
Major concern:
In instruction: the epidemiological history of B subtype influenza in Africa need to be added.
L105: There should be multiple(n≥3) runs of MCMC method combined using LogCombiner. Convergence (i.e., effective sample sizes > 200) of relevant parameters should be assessed using Tracer.
L130 All data should be confirmed, for example, 7560+7595=15155, which is not 15156 as your description.
L259 The role of a deletion of 3 amino acids need to be discussed, the mutation affects the antigenicity? pathogenicity?
Minor concern:
L131: Please explain the first appearance of IQR.
L138: The Adjusted OR (95% CI) of Dyspnea in table 1 was absent. If there is no value, please use '-' to indicate as Table2.
L174: The number of samples per year is not clear in the Figure1.
L185 Figure 1 and 2? Fig.1 is not phylogenetic analyses.
L216 The full name of HPD is not clear.
Author Response
Reviewer 2
This paper introduces the epidemic circulation and cases analysis of influenza B in Senegal from 2010 to 2019. The sample size is abundant. The data is very important for guiding the prevention and control of B subtype influenza for Senegal and Africa. However, the manuscript should be improved as follows.
(AR): Thank you for the great interest you showed for this article, and especially for your insightful comments and suggestions which will undoubtedly help to improve the quality of this paper.
Major concern:
- In instruction: the epidemiological history of B subtype influenza in Africa need to be added.
(AR): As mentioned in the introduction, studies focusing on subtype B in Africa are very few. The rare studies have been done recently in few countries (Cameroon, Kenya or South Africa). This is what motivated the present study whose objective is to describe the epidemiological and genetic dynamics of this influenza subtype in Senegal, in order to help filling this African gap in the context of real influenza vaccination prospects.
- L105: There should be multiple(n≥3) runs of MCMC method combined using LogCombiner. Convergence (i.e., effective sample sizes > 200) of relevant parameters should be assessed using Tracer.
- (AR): Indeed we understand the relevant suggestion of the reviewer. In fact, during our analyses, we first focused on the choice of the model. After choosing the best model, we ran our model using a chain length of 200 million. Then, as suggested by the reviewer, the .log file obtained was assessed using Tracer. It resulted that the Effective Sample Size (ESS) obtained was largely above the arbitrary cut-off of 200 for the parameters. Thus, we are very confident with data obtained using MCMC method.
- L130 All data should be confirmed, for example, 7560+7595=15155, which is not 15156 as your description.
(AR): The mistake is now corrected.
- L259 The role of a deletion of 3 amino acids need to be discussed, the mutation affects the antigenicity? pathogenicity?
(AR): Thank you for this pertinent question. First, in the revised manuscript, we revised the location of deletions (Δ) located at position 190-192 of the HA gene with Δ162-164 after removing signal peptide and to follow international nomenclature of these deletions. B/Victoria lineage viruses from Clade 1A harbored the Δ163-164 (K163 and N164) in the HA1 subunit and defined a new antigenic subgroup represented by B/Norway/2409/2017. These viruses have been detected in several countries (European Centre for Disease Prevention and Control. Influenza virus characterisation – Summary Europe, February 2018. Stockholm: ECDC; 2018. Available from: http://www.ecdc.europa.eu/publications/data/influenza-virus-characterisation-summary-europe-february-2). Δ162-164 (D162, K163 and N164) in B/Vic viruses circulating in Senegal are the major genetic signature of clades 1A.2–1A.4. Based on antigenic tests performed at the Francis Crick Institute influenza WHO-CC of London, it was observed that these isolates were not recognized by some antisera directed against B/Norway/2409/2017 which could impact the efficacy of vaccine-containing B/Colorado/06/2017-like virus, representing the emergent strain of B/Victoria-lineage viruses with Δ163-164 deletion in HA1. However, no particular pathogenicity was reported so far. We added a paragraph discussing these aspects in the revised manuscript (L257-267).
Minor concern:
- L131: Please explain the first appearance of IQR.
(AR): IQR has been explained.
- L138: The Adjusted OR (95% CI) of Dyspnea in table 1 was absent. If there is no value, please use '-' to indicate as Table2.
(AR): Corrected accordingly.
- L174: The number of samples per year is not clear in the Figure1.
(AR): We understand the point of view of the reviewer. Figure 1 aimed at presenting the relative proportions of types and lineages over years and not the exact numbers. We added these proportion for a better understanding.
- L185 Figure 1 and 2? Fig.1 is not phylogenetic analyses.
(AR): Indeed, the mistake was corrected.
- L216 The full name of HPD is not clear.
(AR): HPD has been explained in the new version.